# Differentiating Glioblastomas from Solitary Brain Metastases: An Update on the Current Literature of Advanced Imaging Modalities

**DOI:** 10.3390/cancers13122960

**Published:** 2021-06-13

**Authors:** Austin-John Fordham, Caitlin-Craft Hacherl, Neal Patel, Keri Jones, Brandon Myers, Mickey Abraham, Julian Gendreau

**Affiliations:** 1School of Medicine, Mercer University School of Medicine, Savannah, GA 31404, USA; AustinJohn.Fowler.fordham@live.Mercer.edu (A.-J.F.); Caitlin.Elizabeth.Hacherl@live.mercer.edu (C.C.-H.); neal.atul.patel@live.mercer.edu (N.P.); 2Graduate Medical Education, Eisenhower Army Medical Center, Augusta, GA 30905, USA; keri.l.jones18.ctr@mail.mil (K.J.); brandon.l.myers20.mil@mail.mil (B.M.); 3Department of Neurosurgery, University of California San Diego, La Jolla, CA 92093, USA; mabraham@health.ucsd.edu; 4Department of Biomedical Engineering, Johns Hopkins Whiting School of Engineering, Baltimore, MD 21218, USA

**Keywords:** arterial spin labelling, diffusion tensor imaging, diffusion-weighted imaging, dynamic susceptibility contrast, glioblastoma, intracranial, machine learning, magnetic resonance, metastasis, neurite orientation dispersion density

## Abstract

**Simple Summary:**

The process of differentiating glioblastomas from solitary brain metastases is often difficult using traditional magnetic resonance imaging alone. In the past two decades, much progress has been made in devising advanced imaging modalities for the purpose of ascertaining more data on these intracranial tumors to help neuroradiologists in differentiating the two pathologies. In addition to the data provided by dynamic susceptibility contrast imaging and magnetic resonance spectroscopy, more innovative modalities now include diffusion tensor imaging and neurite orientation dispersion and density imaging. Radiomic analysis protocols and machine learning algorithms are being continually optimized to increase the accuracy of diagnosis by utilizing multiple different imaging protocols per patient. In this review, we provide an update on these advanced imaging modalities by reviewing the most up-to-date and current evidence.

**Abstract:**

Differentiating between glioblastomas and solitary brain metastases proves to be a challenging diagnosis for neuroradiologists, as both present with imaging patterns consisting of peritumoral hyperintensities with similar intratumoral texture on traditional magnetic resonance imaging sequences. Early diagnosis is paramount, as each pathology has completely different methods of clinical assessment. In the past decade, recent developments in advanced imaging modalities enabled providers to acquire a more accurate diagnosis earlier in the patient’s clinical assessment, thus optimizing clinical outcome. Dynamic susceptibility contrast has been optimized for detecting relative cerebral blood flow and relative cerebral blood volume. Diffusion tensor imaging can be used to detect changes in mean diffusivity. Neurite orientation dispersion and density imaging is an innovative modality detecting changes in intracellular volume fraction, isotropic volume fraction, and extracellular volume fraction. Magnetic resonance spectroscopy is able to assist by providing a metabolic descriptor while detecting variable ratios of choline/N-acetylaspartate, choline/creatine, and N-acetylaspartate/creatine. Finally, radiomics and machine learning algorithms have been devised to assist in improving diagnostic accuracy while often utilizing more than one advanced imaging protocol per patient. In this review, we provide an update on all the current evidence regarding the identification and differentiation of glioblastomas from solitary brain metastases.

## 1. Introduction

Gliomas are found to account for approximately 25% of all adult brain tumors. They are considered the most rapidly growing malignancies of the central nervous system, with glioblastoma multiforme comprising more than 50% of all gliomas [1]. In addition, solitary brain metastases are becoming increasingly more common as the elderly population continues to increase in number throughout the world. Additional diagnoses of intracranial metastases are also being made due to the increased use of imaging for clinical assessment. Therefore, differentiating these two diagnoses is becoming a more prominent and challenging issue in patient care for neuroradiologists and neurosurgeons [2,3,4]. In many cases, a biopsy is performed for histological confirmation, even if there is a history of known malignancy [5].

Accurate tumor diagnosis and classification is essential early on in a patient’s clinical assessment, as treatment modalities differ significantly between the two diseases. Patients with glioblastoma typically undergo gross total resection and treatment consisting of both radiotherapy and a chemotherapeutic course of temozolomide [6]. Metastases are commonly managed with stereotactic radiosurgery or open operative resection with postoperative whole-brain radiotherapy depending on the size and location of the disease [7]. This is in addition to the evaluation for the possibility of systemic cancer if the patient has no known prior history. Having the earliest detection and differentiation of the patient’s disease is paramount for optimizing clinical outcomes.

Initially, imaging modalities for discerning these diseases was limited to conventional high-resolution magnetic resonance imaging (MRI) with a contrasting agent such as gadolinium for optimal characterization. However, this imaging is limited, as both pathologies are largely characterized by having peritumoral hyperintensity and FLAIR signal (Figure 1) [8,9,10,11,12,13,14,15]. In the past decade, advanced imaging modalities were developed to augment traditional MRI in improving diagnostic accuracy. To date, an update on these novel imaging techniques and procedures has yet to be provided. Therefore, the goal of this review is to outline the most recent findings of newly developed and tested imaging systems by providing data on clinical trials, case series, and technical reviews. In addition, an overview of the physics and mechanisms of each modality will be discussed.

## 2. Magnetic Resonance Imaging

This technology is fundamental for initial evaluation of intracranial lesions and, additionally, is a basis for many of the advanced imaging modalities used for further characterization. During the imaging process, the patient lies supine and is placed inside powerful magnets producing magnetic fields. These fields align protons of the imaged tissue in the same direction as the magnetic fields in a parallel fashion. A radiofrequency is then applied, stimulating protons to spin out of equilibrium, which causes strain against the pull of the magnetic field. Sensors detect the energy emitted from these protons once this radiofrequency field is turned off and the protons relax [16]. These sensors generate an image at a much higher resolution than computed tomography and other radiographs. This allows for a more superior diagnostic accuracy of many intracranial lesions (Figure 2).

The T1-weighted and T2-weight MRI sequences describe two different relaxation times. T1 (longitudinal relaxation time) is the rate at which excited protons return to equilibrium and realign with the magnetic field. The T2 (transverse relaxation time) is the rate at which the spinning protons lose phase coherence among the nuclei spinning perpendicular to the main field. 

Once the high-resolution MRI image is produced, both glioblastoma and metastases have similar features. They are both surrounded by a T2 signal hyperintensity that has been traditionally termed vasogenic edema. The nonenhancing hypointense area that surrounds the outer aspect of the tumor is peritumoral edema. This signal abnormality represents increased interstitial water due to capillary permeability and breakdown of the blood–brain barrier. Glioblastoma is infiltrative and tends to invade surrounding tissue and white matter tracts. It advances microscopically for several centimeters beyond the area of enhancement on imaging [5]. Therefore, in glioblastoma, peritumoral areas demonstrate not only increased interstitial water but also the possibility of scattered metastases, which is described as infiltrative edema [5]. This is in comparison with metastasis, which grows in an expansile manner, only displacing surrounding tissues and without infiltrative edema. This concept suggests that the most successful methods for accurately characterizing the lesion would focus and assess for peritumoral parameters [17]. Typical MRI has limited ability to detect these peritumoral changes; therefore, novel advanced MRI concepts are being developed to aid in this process (Table 1).

It is important to note that there is some evidence for differentiating the two diseases using conventional MRI alone. One study found that the ratio of the maximal diameter of the peritumoral area to the maximal diameter of the enhancing mass can be used to help distinguish between the two images. A lower ratio favors glioblastoma. Using a cutoff value of 2.35, this achieves an accuracy of 68%, sensitivity of 84%, and specificity of 45% [18]. Relatively, there is increased water in the vasogenic edema of metastases when compared with glioblastoma resulting from the increased neoplastic cells in the latter. Therefore, there would be a theoretical decrease in signal area for glioblastoma when compared with metastases with fluid-attenuated inversion recovery (FLAIR) sequences. This method is suggested to have a sensitivity of 44% and specificity of 91% [10]. Metastases can generally be expected to involve the subcortical white matter and grey–white matter junction.

## 3. Perfusion Magnetic Resonance Imaging

### 3.1. Dynamic Susceptibility Contrast-Enhanced Perfusion

When traditional MR imaging is coupled with contrasting agents such as gadolinium, increased diagnostic accuracy is achieved [27]. Contrast agents are now found in largely 60% of MRI examinations performed for neurological disease [28]. Dynamic susceptibility contrast-enhanced perfusion (DSC) is one of the most widely used methods of MR perfusion imaging due to its efficacy, wide availability, and quick visualization times of <1 min [29,30,31,32,33,34]. A succinct magnetic resonance signal is generated from this gadolinium contrast agent by tracking T2*-weighted signal changes during the first pass of contrast [31]. A map of the cerebral blood volume (CBV) that outlines the vascularity of the tumor is created [15,29]. In some tumors, marked angiogenesis occurs, which produces increased relative peritumoral cerebral blood volume (rCBV) when compared with the surrounding brain parenchyma [35].

Unlike single brain metastases, glioblastoma presents with elevated rCBV due to increased cellular proliferation. Therefore, this allows DSC to be useful as a diagnostic tool [13,36,37]. In metastatic tumors, the peritumoral region is intrinsically normal parenchyma but edematous since metastases do not induce surrounding neoangiogenesis [38,39,40,41,42,43]. This edema is additionally suggested to provide local compression of blood flow, further decreasing rCBV when compared with glioblastoma [44,45]. Park et al. describe an autoencoder used for DSC that is found to be helpful in differentiating the two diseases by using perfusion patterns [46]. Increased rCBV is additionally suggested to be associated with worse clinical outcomes and shorter overall survivability due to its correlation with aggressive tumor growth in glioblastoma [30,47,48]. 

There is sizeable evidence from previous studies that supports the utility of rCBV as a diagnostic tool to differentiate the two lesions clinically. A meta-analysis by Suh et al. provided data that were statistically analyzed from 18 studies and included a total of 900 patients [19]. They found that DSC offered a high sensitivity of 90% (95% CI = 84–94%) and a specificity of 91% (95% CI = 84–95%) of detecting glioblastoma from metastases using rCBV. Neska-Matuszewska et al. found that brain metastases have significantly lower mean rCBVmax values, in addition to rCBV, when compared with glioblastoma [31]. 

Higher rates of inflow-based vascular-space-occupancy arteriolar CBV and relative arteriolar CBV are also found in glioblastoma. The peak height for peritumoral T2 lesions and signal intensity recovery were higher in glioblastoma than metastases [49]. Additional studies have found that optimizing MR protocols for acquiring percentage signal recovery (PSR) in addition DSC protocols with rCBV acquisition leads to even higher amounts of accuracy [50].

Despite its utility and widespread use for providing a discriminator using rCBV parameters, DSC does have some limitations. It has the potential for artifacts such as the presence of susceptibility artifacts from surgical hardware or bone–air interfaces near the skull base [30,34,51,52,53]. Large cortical veins and intraparenchymal hemorrhages can also be the etiology of artifacts [33,54,55,56]. To aid neuroradiologists in the successful application of DSC MRI, many organizations, such as the American Society of Functional Neuroradiology, have set forth guidelines of standardization to attain reproducible and satisfactory results [57]. These recommendations include inspecting signal-time curves for the presence of an appropriate signal drop to indicate a quality contrast bolus [58] and inspecting perfusion maps for bolus or processing errors [53].

### 3.2. Dynamic Contrast-Enhanced Magnetic Resonance Perfusion

Theoretically, contrast does not extravasate from blood vessels in the brain; however, in brain tumors, blood does extravasate out of the vasculature. This blood extravasation is measured by dynamic contrast-enhanced MR perfusion (DCE), which was developed in 1990 [59]. Improvements in data quality have since led to more complex second-generation models that are in use by radiologists today [60]. Similar to DSC, DCE provides information about tissue properties at the microvascular level. However, this is done by capturing serial T1-weighted images before, during, and after injecting a gadolinium-based contrast agent [33].

Following contrast agent injection, the resulting hemodynamic signals are dependent on the T1 relaxation time and increase (T1 shortening) due to the effects of the paramagnetic or gadolinium-based contrast agent [61]. A linear relationship between the magnetic resonance signal and the concentration of the contrast agent in the tissue results. This T1 shortening is the mechanism of tissue enhancement. To measure these signal changes as a function of time, T1-weighted images are captured rapidly and repeatedly [62]. The rate at which the contrast agent diffuses from the blood to the extravascular extracellular space is determined by tissue perfusion, capillary permeability, and capillary surface area [62].

Signal enhancement can be assessed semiquantitatively by analyzing signal intensity changes or quantitatively using pharmacokinetic modeling techniques to determine changes related to contrast agent concentration [62]. The leaking of contrast from the tumor vasculature allows the calculation of quantitative biomarkers, including the transfer constant (Ktrans), the fractional volume of the extravascular–extracellular space (v_e_), the rate constant (Kep), the fractional volume of the plasma space (v_p_), and the area under the curve (AUC) [63]. Ktrans is the biomarker most commonly used and has been shown to indicate permeability in patients with glioma [33,64].

Clinically, this imaging is largely used in conjunction with other imaging modalities for lesion assessment. Jung et al. describe 26 patients with glioblastoma and 32 patients with metastatic brain lesions who underwent DCE imaging. They found that neither Ktrans nor V_p_ was able to differentiate glioblastoma from brain metastases to any statistical significance [65]. However, they were able to differentiate hypervascular lesions, such as glioblastoma and melanoma metastases, from hypovascular metastases using the AUC and the log slope of the washout phase of the signal intensity time curve. Still, these metrics were unable to differentiate glioblastoma from melanoma metastasis. Conversely, Bauer et al. found that Ktrans was higher in patients with glioblastoma than in those with metastatic disease [22]. Zhao et al. found that in the tumor parenchyma, mean V_e_ and initial AUC were significantly higher in primary central nervous system lymphoma and metastases than in high-grade gliomas (*p* < 0.003) [66]. Overall, there seems to be a lack of studies using DCE alone to discriminate between glioblastoma and brain metastasis. Therefore, it is more commonly used as an adjunct in conjunction with other imaging modalities at this time.

With DCE, the ability to quantitatively analyze the blood–brain barrier and vascular permeability gives the user a different perspective compared with DSC and may offer a more complete assessment of brain tumor angiogenesis when used as an adjunct imaging modality [33]. DSC provides the benefit of removing T1 effects, and DCE the benefit of removing T2 * signal change effects when assessing the tumors. 

### 3.3. Arterial Spin Labeling

Arterial spin labeling (ASL) is a noninvasive perfusion imaging technique that uses magnetically labeled arterial blood as a tracer in place of gadolinium-based contrast. This was created in 1992, as it was found to have the benefit of bypassing adverse events of traditional contrast [67,68]. This system works by utilizing the magnetic field of the MRI to magnetize blood just below the region of interest (often in the area of the neck for brain imaging), which is timed to occur at an interval before emitting the pulse frequency. Subsequently, this magnetized blood reaches the brain during the time of the pulse frequency and is already premagnetized. Therefore, the longitudinal magnetization of the blood flow is less than fully relaxed when compared with the stationary tissue. This results in a noninvasive perfusion image [69].

Several studies have shown that cerebral blood flow (CBF) with this modality offers accurate correlation with both CBV and actual vascular density from histological examination even without the use of traditional contrast [68,70,71,72,73]. It has also been found to be sensitive to changes in tumor perfusion specifically [74]. To date, there is indeed evidence concluding that noninvasive ASL techniques can effectively differentiate glioblastoma from primary CNS lymphomas and radionecrosis of the brain [75,76,77]. Direct studies using ASL methods to differentiate glioblastoma from metastases using signal intensity have shown diagnostic benefit; however, ASL-derived tumor blood flow is largely inconclusive on whether it reliably differentiates [78,79,80] in glioblastoma versus metastases specifically. ASL was found to reliably differentiate when coupled with DTI methods in one study [79].

## 4. Diffusion Imaging

### 4.1. Diffusion-Weighted Imaging—Measurement of Apparent Diffusion Coefficient

Diffusion-weighted imaging (DWI) was first introduced in 1984 as a form of conventional MRI that measures the diffusion rate of water within a given tissue without requiring extra equipment, contrast agents, or chemical tracers [81]. This method is inherently limited due to its inability to describe the degree of anisotropy or quantify information regarding structural orientation [82]. DWI is utilized to grade and differentiate tumors based on cellularity [83].

Water diffusion is based on the random Brownian motion of molecules due to thermal energy. In a completely uniform medium, water diffusion is said to be isotropic (the same in all directions), but in a multifaceted system, such as the human body, compartmentalization of water creates anisotropic (directional) diffusion [84]. When a temporary magnetic field is applied, the homogeneity of molecules can be dephased and then followed by a gradient in the opposite direction to rephase and restore signal. The protons that move between the dephasing and rephasing gradients contribute to the diffusion-weighted image [85]. The apparent diffusion coefficient (ADC) describes the magnitude of water diffusion in tissue [86]. With multiple DW images, ADC values can be determined by applying theoretical mathematical equations using variables such as the strength of the magnetic field along with initial signal intensity and signal intensity post-imaging [85].

The images obtained from DWI alone yield little diagnostic value in differentiating gliomas from metastases (Figure 3) [87,88,89,90]. However, one study provides evidence that tumor ADC values in malignant glioma are different from those in metastases by using 3 Tesla MRI equipment at a level that achieves statistical significance [11,91]. Texture analysis with ADC is also found to be a reliable differentiator [92]. Theoretically, models could be developed to examine the cellularity of intracranial lesions using DWI, and it would aid in differentiation. It has been suggested that a reduction in ADC values on imaging indicates higher cellularity, which could be useful in measuring whether or not there is tumor cellular invasion of the surrounding tissues. Several studies have studied this, comparing the peritumoral edema of metastases and high-grade glioma, and support this hypothesis [11,88,88,93,94,95].

### 4.2. Diffusion-Weighted Imaging—Exponential Measurements

DWI signal intensity is actually influenced by intrinsic T2 properties of the brain parenchyma in addition to water diffusibility. Relative signal intensities from each source are difficult to characterize on DWI alone; therefore, the hyperintense signal on imaging is largely attributed to intrinsic T2 signal and is termed T2 “shine- through.” The negative exponential value of ADC can be calculated using mathematical calculations of the workstation, and the T2 signal intensity can be removed from the tissue being imaged, thus creating a more accurate depiction of diffusion. This allows neuroradiologists to better characterize the tumor. Glioblastoma and metastases theoretically have different signal intensities on exponential DWI. Metastases display decreased signal intensity due to vasogenic edema, while glioblastoma displays intermediate signal intensity when compared with brain parenchyma [96,97].

### 4.3. Diffusion Tensor Imaging

The dataset obtained from DWI can be utilized by an additional technique known as diffusion tensor imaging (DTI) [34]. This type of modeling, first described in 1994 [82,98], provides a method for indirectly measuring the complex neuroanatomical structure of white matter tracts in the brain and spinal cord by determining fiber tract orientation and is often color-coded on imaging (Figure 4) [99]. The direction of water molecule diffusion is acquired mathematically through the use of a 3 × 3 symmetric matrix, which can be expressed by its associated eigenvalues and eigenvectors [100]. The matrix represents an ellipsoid shape, which has been shown to be the best orientation for obtaining the most realistic simulation of water diffusion in white matter fiber tracts [101]. This elliptical depiction can be described as a type of tensor [102].

While DWI can be used to determine ADC values, this model is basic, as it assumes isotropic water diffusion (i.e., the same in all directions) [101]. In reality, water diffusion within white matter changes depending on the direction of measurement. Elements that have an effect on water diffusion in the white matter of the brain include myelin sheaths, cell membranes, structural directions, surrounding tissue types, and the level of permeability of these structures [99]. This means that water does not diffuse equally in all directions within the brain tissue, in the same way that a spherical drop of water in microgravity would. Instead, the water within white matter tracts diffuses in a certain direction, bound by these barriers.

With this in mind, it is easier to understand why an elliptical tensor model elongated parallel to white matter tracts is a more accurate diffusion model than a sphere or cube. It is also important to note that the tensor in this context is often referred to as a voxel. A voxel is the volume contained within a specific region of focus. Due to the tensor representing a three-dimensional (3-D) matrix, the voxel is a 3-D point in space containing all elements (in this scenario, the elements consist of brain tissue) of that space, and this is where the measurements are taken [103]. Analysis of each voxel can provide information on the tissue microstructure contained within.

The parameters given through arithmetical analysis of each tensor model include the mean diffusivity (MD), the fractional anisotropy (FA), and the fiber orientation mapping (tractography) [104]. FA measures the magnitude of the tensor’s associated eigenvalues that can be attributed to anisotropic diffusion, which is equivalent to the direction of water diffusion [105]. In general, FA decreases in damaged tissues due to the disruption of directional water diffusion. The eigenvector correlated with the tensor model can be used to determine the orientation of tissue structure within the brain. This allows the connectivity of white matter fiber tracts to be mapped out voxel by voxel [106]. MD is a measure of the overall diffusion within a voxel. In damaged tissues, MD commonly increases secondary to increased permeability of the barriers to diffusion, such as the cell membrane and myelin sheath [107]. However, as with FA, this is not always the case and can depend on the specific region of the brain, disease pathology, and developmental conditions [108]. In the brain, FA is also a measure of white matter fiber integrity in addition to directionality [109,110]. FA is found to correlate with cell density, axonal structures, neuronal structures, vascularity, and fiber tracts [111,112,113]. It is still largely controversial whether increases in FA values are associated with tumors or vice versa [114,115]. 

Investigators have found differences in infiltrative edema of glioblastoma versus vasogenic edema of metastases [116,117,118]. There is largely higher MD in the peritumoral edema than in infiltrative edema. This is due to the marked influx of water when compared with the cellular infiltrate of glioblastoma. Peritumoral FA is not significantly different between the two pathologies and is conflicting, likely due to the infiltrative edema having varying fractional compositions of normal white–grey matter, intracellular water, and tumor cellular infiltrates [5]. 

Jiang et al. performed a meta-analysis including a total of 9 studies with 193 patients having high-grade glioma and 141 patients with metastases [119]. They found a significant increase in FA of the peritumoral region in high-grade glioma when compared with single brain metastases (SMD = 0.47; 95% CI, 0.22–0.71; *p* < 0.01). A significant decrease in MD of the peritumoral region was also found in high-grade glioma when compared with metastases (SMD = −1.49; 95% CI, −1.91 to −1.06; *p* < 0.01). Interestingly, no significant differences were detected in FA or MD within the intratumoral area of gliomas and metastasis.

In a more recent and much larger meta-analysis, Suh et al. included 14 studies and a total of 1143 patients to determine the diagnostic performance of DWI and DTI in differentiating high-grade glioma from metastases [20]. They found the average sensitivity and specificity of DWI and DTI in diagnosing high-grade gliomas from solitary metastatic lesions to be 79.8% (95% CI, 70.9–86.4%) and 80.9% (95% CI, 75.1–85.5%), respectively. Peritumoral heterogeneity on texture analysis has also been suggested to differentiate these two diseases in DTI; however, intratumoral texture does not differentiate [120]. When coupled with DSC, DTI parameters were found to achieve an accuracy of up to 98% in one study [22]. In addition, Chen et al. propose a Bayesian-network-based decision support system to differentiate glioblastomas from solitary metastases using DTI, DSC, and FLAIR sequences [121].

### 4.4. Neurite Orientation Dispersion and Density Imaging

Just as DTI is an extension of DWI, providing more details about neuroanatomical microstructure, neurite orientation dispersion and density imaging (NODDI) takes it a step further. The morphology quantification of dendrites and axons, jointly known as neurites, has previously been restricted primarily to postmortem histology [122]. First introduced in 2012, NODDI is a practical diffusion MRI technique that can be used to estimate the intricacy of neurites in vivo on standard MRI scanners [123]. Neurite density and dispersion within the brain tissue can be mapped out and provide helpful information on brain connectivity [124]. For example, neurite density decreases as the brain ages [125], as well as in neurodegenerative disorders, such as multiple sclerosis [126], amyotrophic lateral sclerosis [127], and Alzheimer’s disease [128]. The severity of disease progression can be assessed through the quantification of neurite density by NODDI analysis. In the same way that neurite density provides evidence of neurodegenerative disease, it can also provide data to illuminate other disease pathology, such as gliomas or brain metastasis [21]. 

NODDI simplifies the brain architecture into three compartments per voxel: the intracellular space, extracellular space, and cerebrospinal fluid. In this way, NODDI can provide more specific details on the microstructural changes of neurites than DTI analysis alone [123]. While DTI measures indices, such as FA and MD, to map out water diffusion within regions of interest, NODDI creates a compartment map by generating intracellular volume fraction (VIC), isotropic volume fraction (VISO), and extracellular volume fraction (VEC) [21].

In a small study of 15 patients (9 with confirmed glioblastoma and 6 with confirmed metastases), Kadota et al. found that VEC in the peritumoral signal change area was most useful in differentiating glioblastoma from metastases when compared with FA, MD, VIC, and VISO [21]. They determined that VEC provided 100% sensitivity and 83.3% specificity with a threshold value ≥0.48. Inherent limitations of this study include the small sample size and not evaluating peritumoral T2-hyperintense regions.

In another recent study, Mao et al. compared the results of five diffusion-weighted MRI models in differentiating high-grade glioma from metastases [129]. This included a total of 42 previously untreated patients, 20 with confirmed high-grade glioma and 21 with confirmed metastases. They found that NODDI outperformed DTI and DWI in differentiating between high-grade glioma and metastases [130]. The single best parameter for differentiation between the two was found to be VISO. It is important to note that, unlike in the previous study mentioned, Mao et al. did not measure VEC [129].

## 5. Metabolic Imaging

### 5.1. Magnetic Resonance Spectroscopy 

Magnetic resonance spectroscopy (MRS) is a unique modality from the other commonly used imaging techniques in that it provides data on multiple metabolic parameters. This modality has the ability to detect N-acetylaspartate (NAA), creatine (Cr), choline (Cho), glutamine, glutamate, myoinositol, lipid, lactate, and γ-aminobutyric acid (GABA). MRS is based primarily on the detection of signal arising from hydrogen nuclei; however, other nuclei can also arise with signal. These nuclei consist of a nonzero intrinsic magnetic moment, which can be manipulated by magnetic fields. These intrinsic moments arise from the spin of the nuclei, which is an effect of the quantum mechanical spin of the protons and neutrons of the nucleus [131]. From these spins, magnetization generates a magnetic resonance spectrum from these hydrogen protons. Frequencies are then detected by a radiofrequency coil placed near the brain. The electromagnetic force that is generated is proportional to the magnitude of magnetization, and this proportion allows the MRS to provide spectra of the signal source [131]. MRS is a magnetic resonance spectrum that graphically displays detected signals as a function of temporal frequencies (Figure 5).

Initial use of MRS for the detection and differentiation of brain tumors was largely limited as it showed great signal scatter and generally overlapped with normal brain tissue [132]. However, many improvements and developments have taken place, allowing radiologists to discern specific variables from these tumors used for identification. For identification purposes, prominent lipid signal indicates general cellular necrosis in both glioblastomas and metastases. However, new data show that MRS absolute lipid and macromolecular signals could help in differentiating glioblastoma from metastases. The LM13 class specifically was found to be a discriminant parameter being elevated with metastasis found to have an accuracy of 85% [23]. Lipid peak area ratios were also found to be 2.6 in glioblastoma when compared with 3.8 in metastasis when using short echo time 1H spectroscopy [133].

Peritumoral NAA/Cr, Cho/ Cr, and Cho/NAA ratios also have evidence to support their use to differentiate glioblastoma from intracranial metastases at 3 Tesla [134,135]. Cho increases in the presence of increased cellular membrane turnover and proliferation; thus, it would theoretically be elevated in areas of tumor infiltration. This is true even in the absence of enhancement of T2 signal abnormality [15,136]. The highest Cho compounds are found in the peritumoral area of glioblastoma when compared with metastasis overall [120]. 

In addition, the presence of intratumoral Cr suggests glioma, whereas the absence of Cr indicates metastasis [137]. Choline/Cr greater than 2.48 is useful in detecting metastasis and glioblastoma when compared with <2.48, which would be indicative of cerebral infarction or radiation necrosis [138]. Intratumoral Cho/Cr ratio is not as useful as peritumoral Cho/Cr in differentiating the two diseases [139,140,141,142,143,144]. 

Deep learning protocols for MRS are being utilized to better characterize tumor disease. These protocols utilize complex nonlinear relationships and patterns found in both glioblastoma and metastases and identify a metabolic “footprint” instead of simply quantifying metabolite concentrations in the tumor [121,145].

### 5.2. Positron Emission Tomography

This imaging modality utilizes positron-emitting radiopharmaceuticals, such as (18) F-fluorodeoxyglucose (for imaging glucose metabolism) [146]. This imaging is derived from a process where decay occurs by the conversion of a proton into a neutron with positron emission. When the emitted proton unites with a nearby electron in an annihilation event, a pair of two photons (γ-rays) are produced, each with an energy of 511 keV traveling in opposite directions. The γ-rays are detected by sensors on the positron emission tomography (PET) scanner. When both photons are detected in one coincidental event, an image is produced and is assumed to have occurred along the line of response (Figure 6) [147].

PET imaging using α [11C] methyl-l-tryptophan was found to differentiate glioblastoma from brain metastases and has an accuracy of 74%. A kinetic tracer analysis separating tumoral tryptophan transport from unidirectional uptake rates using dynamic imaging was found to have a predictive value of 82%. The combined accuracy of the two was 93% [24]. Metabolic uptake, however, is similar between glioblastoma and metastases but not different from any statistical significance [148]. One new advancement in PET imaging is using amino acids such as O-(2-[^18^F]fluroethyl)-L-tyrosine (^18^F-FET) to allow for better characterizations of glioma. This achieves better characterization than PET imaging with glucose-based tracers, as glucose-based tracers have poorer contrast with the surrounding brain parenchyma. Amino acids have greater contrast as the neoplastic tissue uptake is clearly elevated since it has higher protein metabolism than the surrounding tissue. This allows for improved differentiation of glioma from metastasis [149]. 

### 5.3. Single-Photon Emission Computed Tomography 

Single-photon emission computed tomography (SPECT) has a mechanism of action similar to PET imaging; however, it requires a single physical collimator that absorbs a large fraction of the photons. Therefore, PET imaging is more sensitive than SPECT by two orders of magnitude [150].

When considering its usefulness for differentiating between glioblastomas and metastases, however, there is little evidence supporting the utility of this system. SPECT is useful for detecting bodywide metastases from glioblastoma [151].

## 6. Phase Difference-Enhanced Imaging

A new advanced imaging modality is able to enhance both paramagnetic and diamagnetic substances while being able to select which phases are enhanced. In addition, it can selectively enhance the contrast between the target tissue and the surrounding tissue. This is termed phase difference-enhanced imaging (PADRE). Doishita et al. developed color PADRE, which enables simultaneous visualization of myelin-rich structures and veins. They tested their new method on 11 patients. They found that the visibility of the superficial white matter was increased in metastases when compared with glioblastoma. Therefore, assessment of peritumoral areas with PADRE does appear to have the ability to assist with differentiation at this time [152].

## 7. Radiomics-Based Machine Learning

One innovative area of radiology is within machine learning, using computerized algorithms and protocols to achieve superior accuracy and sensitivity of imaging techniques. The advent of machine learning has resulted in its algorithms being increasingly used for a number of pathologies. These systems utilize support vector machines (SVMs) and multilayer perceptrons (MLPs) for the classification of brain tumors [121,153,154,155,156]. These are largely performed over several MR sequences. Radiomics analysis uses extracted data to include the patients’ clinical information, tumor location, morphology, and wavelet features to a high degree of accuracy to assist with machine learning (Figure 7) [157,158,159,160].

Swinburne et al. found that when using machine learning strategies with a variety of MR-specific sequences, including DWI, DCE, and DSC, machine learning capabilities can differentiate between glioblastomas and metastases with modest diagnostic accuracy. It has been found to provide up to 19% increase in diagnostic yield when added to human interpretation [161,162]. Qian et al. further found that one of their radiomic machine learning classifiers was superior in successfully differentiating glioblastoma from metastasis when compared with neuroradiologist interpretation in 412 patients [160]. Texture features are more significant than fractal-based features in differentiating the two diseases [163]. Three-dimensional morphometric analysis has been used to provide up to 95.8% accuracy when using two three-dimensional shapes per tumor for classification. These three-dimensional features can be objectively derived from quantitative imaging [164].

Tsolaki et al. propose a machine learning algorithm that combines MRS data with DSC in the peritumoral area, leading to an accuracy of up to 98% [25]. Yang et al. developed a computerized system using a semiautomatic segmentation method for DTI imaging and two-dimensional morphological feature extraction and selection and a pattern recognition module for tumor classification. This achieved an accuracy of 97.9% [26]. Therefore, machine learning algorithms have great potential to become optimized and improve diagnostic accuracy in the future. This is especially true when they incorporate multiple, different advanced imaging modalities that could potentially be used for differentiation (Table 1).

Additionally, molecular characterization of these tumors has advanced considerably in the past decade. There is a distinction between IDH1 mutant tumors and EGFR mutant glioblastomas. While tumors with IDH1 mutations often have a more favorable prognosis, EGFR mutations suggest more aggressive tumors. IDH1 has been found to create a hypermethylator phenotype, while EGFR mutations act as functional driver genes [165,166]. Rathore et al. found that MRI could be successfully used to differentiate between these two mutations using radiomic MRI signatures. They were able to differentiate a total of three different phenotypic subtypes by MRI [167]. Perfusion signatures have also been found to be accurate in detecting EGFRvIII status [168].

## 8. Conclusions

Conventional MRI is standard for initial workup of intracranial tumors; however, it remains difficult to differentiate the two pathologies when using conventional MRI as the sole imaging modality. Optimal parameters for the use of rCBV in DSC is becoming much more delineated. New advancements have been made with diffusion imaging techniques to include DTI, in addition to NODDI, which appears to be promising for accurate differentiation in the future. Finally, multiple algorithms for radiomics and machine-based learning programs are being developed, often incorporating multiple advanced imaging modalities to achieve high levels of accuracy.

## Figures and Tables

**Figure 1 cancers-13-02960-f001:**
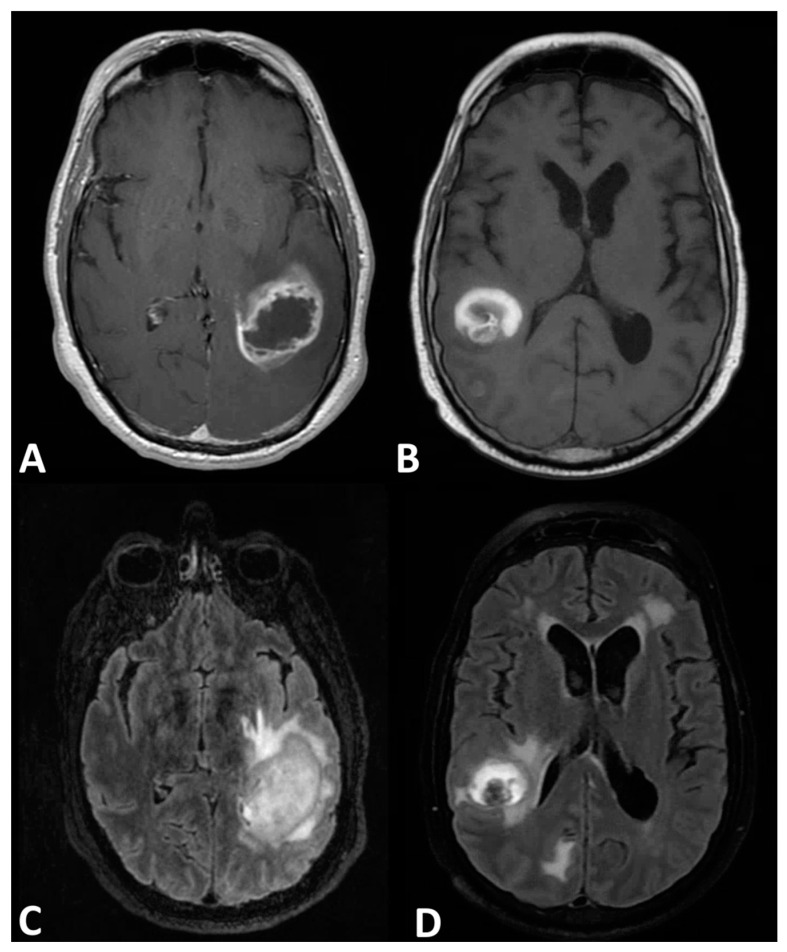
Magnetic resonance imaging. T1 imaging of a left temporal–parietal (**A**) glioblastoma and a left temporal–parietal (**B**) solitary intracranial metastasis both exhibiting ring-enhancing hyperintensity features with peritumoral hypointensity. T2 imaging of (**C**) a glioblastoma and a (**D**) solitary metastasis both characterized by peritumoral hyperintensity.

**Figure 2 cancers-13-02960-f002:**
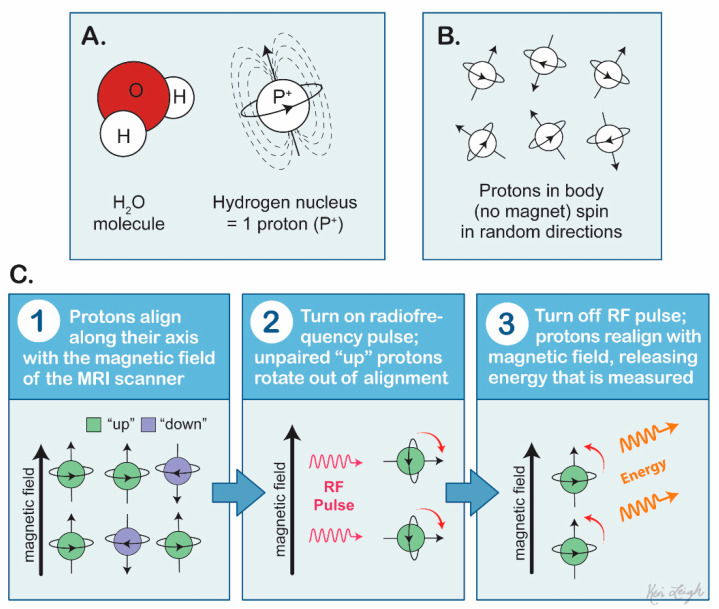
Mechanism of magnetic resonance imaging. (**A**) Hydrogen atoms are dispersed within the patient’s tissue with intrinsic spin. (**B**) Hydrogen atoms are spinning in random direction without alignment with one another. (**C**) Protons align with the magnetic field in parallel fashion; after the application of a radiofrequency pulse, the protons realign with the magnetic field, releasing energy and generating a high-resolution image of the tissue.

**Figure 3 cancers-13-02960-f003:**
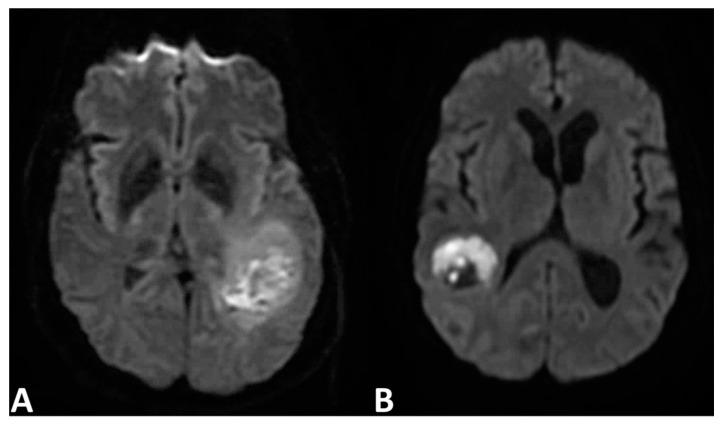
Diffusion-weighted imaging of (**A**) glioblastoma and (**B**) solitary intracranial metastasis with largely similar imaging features.

**Figure 4 cancers-13-02960-f004:**
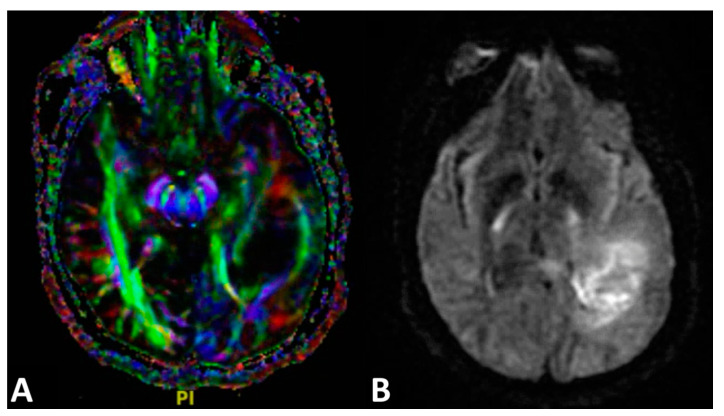
Diffusion tensor imaging. (**A**) Colored diffusion tensor imaging of a left temporal–parietal glioblastoma. Different neural fibers are displayed: transverse fibers (red), anteroposterior fibers (green), and craniocaudal fibers (blue) using tensor imaging features. (**B**) Noncolored diffusion tensor imaging of a glioblastoma.

**Figure 5 cancers-13-02960-f005:**
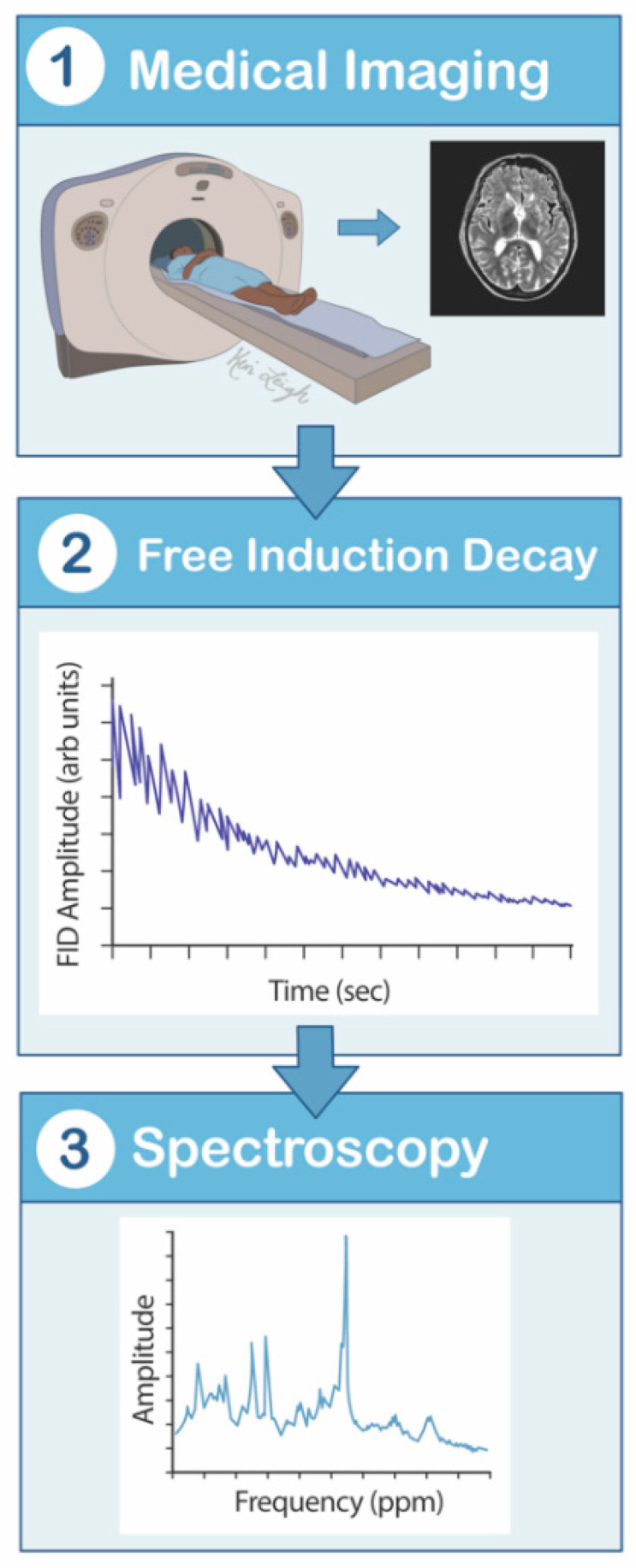
Magnetic resonance spectroscopy mechanism flowsheet. (1) The patient lies supine in the MRS scanner. (2) The hydrogen nuclei contain a nonzero intrinsic quantum mechanical spin of the protons and neutrons, which is manipulated by magnetic fields. Magnetization is applied, creating an observed nuclear magnetic resonance signal. Frequencies are then detected by a radiofrequency coil placed near the brain, creating a free induction decay. (3) The electromagnetic force that is generated is proportional to the magnitude of magnetization, and this proportion allows the MRS to provide spectra of the signal source. MRS is a magnetic resonance spectrum that graphically displays detected signals as a function of temporal frequencies. MRS—magnetic resonance spectroscopy.

**Figure 6 cancers-13-02960-f006:**
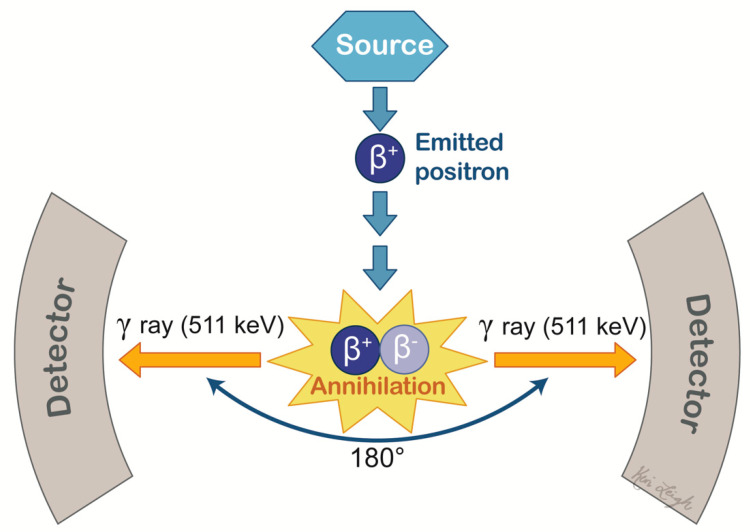
Positron emission tomography scan mechanism. Decay occurs by the conversion of a proton into a neutron with positron emission. When the emitted positron unites with a nearby electron in an annihilation event, a pair of two photons (γ-rays) at 511 keV are produced, each traveling in opposite directions. The γ-rays are detected by sensors on the PET scanner. When both photons are detected in one coincidental event, an image is produced and is assumed to have occurred along the line of response.

**Figure 7 cancers-13-02960-f007:**
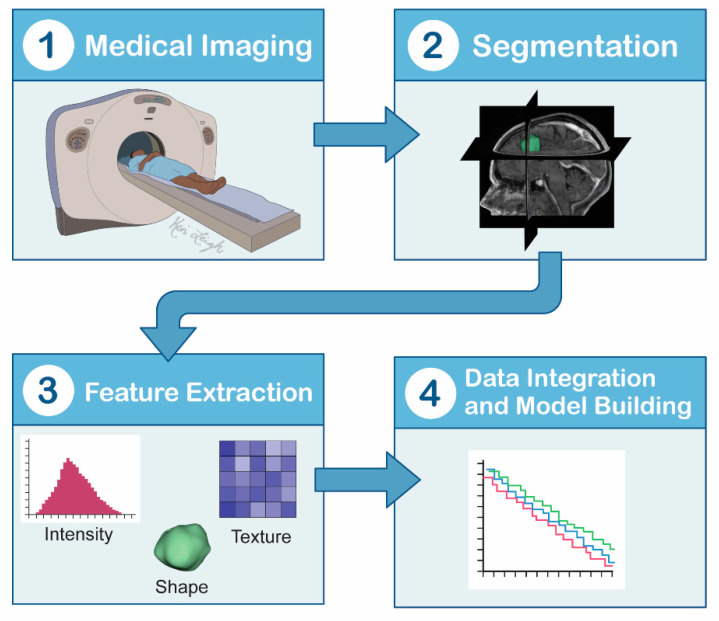
Radiomics and machine flow learning. (1) A medical image is generated using any one of the advanced imaging modalities or multiple, different modalities. (2,3) Segmentation of the imaging is performed while also extracting feature data, such as intensity, texture, and shape. (4) This allows radiologists to build programs and algorithms to increase the sensitivity of imaging by creating systems that utilize support vector machines and multilayer perceptrons.

**Table 1 cancers-13-02960-t001:** Sensitivities and specificities of advanced imaging modalities for differentiating glioblastoma versus metastasis.

Imaging Technique	Sensitivity	Specificity	Accuracy
Traditional MRI with ratio of peritumoral area to enhancing mass with a cutoff value of 2.35 [18]	84%	45%	68%
Decrease in FLAIR signal in glioblastoma compared with metastasis [10]	44%	91%	
Dynamic susceptibility contrast perfusion using rCBV [19]	90%	91%	
Both DWI and DTI [20]	79.8%	80.9%	
Diffusion tensor imaging using a VEC threshold of 0.48 [21]	100%	83.3%	
DTI parameters with DSC [22]	60–91%	55–100%	98%
MRS with LM13 class lipids and cutoff of 81 mM [23]	81%	78%	85%
PET imaging with α [11C] methyl-l-tryptophan with kinetic tracer analysis [24]			93%
Machine learning algorithm with MRS and DSC data [25]			98%
Two-dimensional morphological feature extraction for DTI [26]			97.9%

FLAIR—fluid-attenuated inversion recovery; rCBV—relative cerebral blood volume; VEC—extracellular volume fraction; DWI—diffusion-weighted imaging; DTI—diffusion tensor imaging; DSC—dynamic susceptibility contrast; MRS—magnetic resonance spectroscopy; PET—positron emission tomography.

## Data Availability

Not applicable.

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
