# Peer review of "Differentiating Glioblastomas from Solitary Brain Metastases: An Update on the Current Literature of Advanced Imaging Modalities"

_cancers, 2021, doi:10.3390/cancers13122960_

Round 1

Reviewer 1 Report

I read with great pleasure the review by Fordham and Colleagues, that aims at elucidating whether it is nowadays possible and clinically relevant to distinguish between glioblastoma and brain metastases through different neuroimaging techniques. The review is written in a clear, simple and straightforward manner and summarizes the literature highlighting the major features of each single technique. I believe that this Review is helpful for neuroradiologists and residents, but may also be of benefit to neurologists and oncologists because it provides an up-to-date and comprehensive glimpse at the available evidence in the field. 

I have only some minor requests to improve readability, concerning some text mistakes:

  1. line 55: 'differentiating...IS becoming...'
  2. line 66: '...no knowN...'
  3. line 86: '...a glioblastoma [delete 'characterized] and a...'
  4. line 311: '...can BE removed...'
  5. line 318: 'Tensor Imaging' in Italics
  6. line 556: '...by creating systems THAT utilize...'

Author Response

I read with great pleasure the review by Fordham and Colleagues, that aims at elucidating whether it is nowadays possible and clinically relevant to distinguish between glioblastoma and brain metastases through different neuroimaging techniques. The review is written in a clear, simple and straightforward manner and summarizes the literature highlighting the major features of each single technique. I believe that this Review is helpful for neuroradiologists and residents, but may also be of benefit to neurologists and oncologists because it provides an up-to-date and comprehensive glimpse at the available evidence in the field. 

Thank you for the kind words.

I have only some minor requests to improve readability, concerning some text mistakes:

  1. line 55: 'differentiating...IS becoming...'
  2. line 66: '...no knowN...'
  3. line 86: '...a glioblastoma [delete 'characterized] and a...'
  4. line 311: '...can BE removed...'
  5. line 318: 'Tensor Imaging' in Italics
  6. line 556: '...by creating systems THAT utilize...'

All of these above edits have been made to the manuscript.

Reviewer 2 Report

This review paper by Fordham et al sets out to discuss advances in the literature to differentiate between glioblastoma and brain metastasis using imaging techniques. This is an important topic. The paper is written clearly and there is good consideration of key concepts in radiology including principles of MRI and DTI for example. However, further consideration should be given to the latest advances in the field, particularly to the molecular characterization of these tumours (which has advanced considerably in the last decade), and how radiology can contribute to discerning the molecular details of the tumour earlier than through biopsy pathology. The following papers, as a minimum, should be discussed and linked into the review:

  • For GBM, there is a key division between IDH1 mutant tumours and EGFR mutant tumours. IDH1 mutation confers a better prognosis, whereas EGFR mutation suggests a poor prognosis; in particular EGFRvIII mutations suggest s an aggressive GBM and indeed EGFRvIII initiates gliomas in mice. These points need to be discussed; please see and refer to these papers for example: Turcan et al 2012 Nature (IDH1 creates hypermethylator phenotype); Noorani et al 2020 Genome Biology (EGFRvIII cooperative driver landscapes)
  • There is evidence of radiomic MRI signatures to differentiate between IDH1-mutant and EGFRvIII-mutant GBMs in patients - see Rathore et al 2018 Scientific reports.
  • There are potential perfusion MRI signatures of EGFRvIII that can be detected in pre-operative imaging to help with establishing GBM prognosis - see Bakas et al 2017 Clinical Cancer Research (in vivo detection of EGFRvIII in GBM via perfusion MRI).

Author Response

This review paper by Fordham et al sets out to discuss advances in the literature to differentiate between glioblastoma and brain metastasis using imaging techniques. This is an important topic. The paper is written clearly and there is good consideration of key concepts in radiology including principles of MRI and DTI for example. However, further consideration should be given to the latest advances in the field, particularly to the molecular characterization of these tumours (which has advanced considerably in the last decade), and how radiology can contribute to discerning the molecular details of the tumour earlier than through biopsy pathology. The following papers, as a minimum, should be discussed and linked into the review:

Thank you for the kind words.

  • For GBM, there is a key division between IDH1 mutant tumours and EGFR mutant tumours. IDH1 mutation confers a better prognosis, whereas EGFR mutation suggests a poor prognosis; in particular EGFRvIII mutations suggest s an aggressive GBM and indeed EGFRvIII initiates gliomas in mice. These points need to be discussed; please see and refer to these papers for example: Turcan et al 2012 Nature (IDH1 creates hypermethylator phenotype); Noorani et al 2020 Genome Biology (EGFRvIII cooperative driver landscapes)
  • There is evidence of radiomic MRI signatures to differentiate between IDH1-mutant and EGFRvIII-mutant GBMs in patients - see Rathore et al 2018 Scientific reports.
  • There are potential perfusion MRI signatures of EGFRvIII that can be detected in pre-operative imaging to help with establishing GBM prognosis - see Bakas et al 2017 Clinical Cancer Research (in vivo detection of EGFRvIII in GBM via perfusion MRI).

All suggested citations have been added to the manuscript.

A paragraph was added to the radiomics section discussing all points.

"Additionally, molecular characterization of these tumors has advanced considerably in the past decade. There is a distinction between IDH1 mutant tumors and EGFR mutant glioblastomas. While tumors with IDH1 mutations often have a more favorable prognosis, EGFR mutations suggest more aggressive tumors. IDH1 has been found to create a hypermethylator phenotype while EGFR mutations act as functional driver genes.164,165 Rathore et al found that MRI could be successfully used to differentiate between these two mutations using radiomic MRI signatures. They were able to differentiate a total of three different phenotypic subtypes by MRI.166 Perfusion signatures have also been found to be accurate in detecting EGFRvIII status.167"

Reviewer 3 Report

In this review the authors have provided in depth review of various MRI based and other techniques that are being used and upcoming to help differentiate GBM vs brain metastases. 

The review is very well written. The flow of the manuscript is good. 2 comments:

1) A table highlighting each of these advanced techniques with their available sensitivities vs specificities 

2) There are other metabolic PET imaging such as FET PET and amino acid PET that are also emerging and can be discussed in the PET paragraph 

Author Response

In this review the authors have provided in depth review of various MRI based and other techniques that are being used and upcoming to help differentiate GBM vs brain metastases. 

The review is very well written. The flow of the manuscript is good. 2 comments:

Thank you for the kind words.

1) A table highlighting each of these advanced techniques with their available sensitivities vs specificities 

Table 1 was included for sensitivities and specificities of all modalities.

2) There are other metabolic PET imaging such as FET PET and amino acid PET that are also emerging and can be discussed in the PET paragraph 

"One new advancement in PET imaging is using amino acids such as O-(2-[18F]fluroethyl)-L-tyrosine (18F-FET) to allow for better characterizations of glioma. This achieves better characterization than PET imaging with glucose-based tracers, as glucose-based tracers have poorer contrast with the surrounding brain parenchyma. Amino acids have greater contrast as the neoplastic tissue uptake is clearly elevated since it has a higher protein metabolism than the surrounding tissue. This allows for improved differentiation of glioma from metastasis.147 "

Round 2

Reviewer 2 Report

The authors have addressed my comments.